# The Negative Impact of Varicocele on Basic Semen Parameters, Sperm Nuclear DNA Dispersion and Oxidation-Reduction Potential in Semen

**DOI:** 10.3390/ijerph18115977

**Published:** 2021-06-02

**Authors:** Kamil Gill, Michal Kups, Patryk Harasny, Tomasz Machalowski, Marta Grabowska, Mariusz Lukaszuk, Marcin Matuszewski, Ewa Duchnik, Monika Fraczek, Maciej Kurpisz, Malgorzata Piasecka

**Affiliations:** 1Department of Histology and Developmental Biology, Pomeranian Medical University, 71-210 Szczecin, Poland; kamil.gill@pum.edu.pl (K.G.); michalkups1@gmail.com (M.K.); patryk.harasny@gmail.com (P.H.); tomasz.machalowski@poczta.onet.pl (T.M.); martag@pum.edu.pl (M.G.); 2TFP Fertility Vitrolive in Szczecin, 70-483 Szczecin, Poland; 3Department of Urology and Oncological Urology, Regional Specialist Hospital in Szczecin, 71-455 Szczecin, Poland; 4Department of Urology and Urological Oncology, Pomeranian Medical University in Szczecin, 70-111 Szczecin, Poland; 5Department of Perinatology, Obstetrics and Gynecology, Pomeranian Medical University, 72-010 Police, Poland; 6Healthcare Center Nowe Orlowo, 81-525 Gdynia, Poland; m.lukaszuk@gumed.edu.pl; 7Invicta Fertility Clinic, 80-850 Gdansk, Poland; 8Department of Urology, Medical University in Gdansk, 80-214 Gdansk, Poland; marcin.matuszewski@gumed.edu.pl; 9Department of Aesthetic Dermatology, Pomeranian Medical University in Szczecin, 70-111 Szczecin, Poland; educhnik@pum.edu.pl; 10Institute of Human Genetics, Polish Academy of Sciences, 60-479 Poznan, Poland; monika.fraczek@igcz.poznan.pl (M.F.); maciej.kurpisz@igcz.poznan.pl (M.K.)

**Keywords:** varicocele, infertility, sperm DNA fragmentation, oxidative stress

## Abstract

Since varicocele is so common in infertile men, this study intends to analyse the relationships between varicocele and conventional semen characteristics, sperm nuclear DNA dispersion and oxidation-reduction potential (ORP) in semen. Varicocele-positive and varicocele-negative infertile men (study groups) showed significantly lower standard sperm parameters and higher sperm DNA fragmentation (SDF) and ORP in semen than healthy volunteers and subjects with proven fertility (control groups). A lower proportion of low SDF levels (0–15% SDF) and higher incidence of high SDF levels (>30% SDF), as well as a higher prevalence of high ORP values (>1.37 mV/10^6^ sperm/mL), were found in the study groups vs. the control groups. Moreover, infertile men had significantly lower odds ratios (ORs) for low SDF levels and significantly higher ORs for high SDF levels and high ORP. SDF and ORP were negatively correlated with sperm number, morphology, motility and vitality. Furthermore, a significant positive correlation was found between SDF and ORP. The obtained results suggest that disorders of spermatogenesis may occur in varicocele-related infertility. These abnormalities are manifested not only by reduced standard semen parameters but also by decreased sperm DNA integrity and simultaneously increased oxidative stress in semen.

## 1. Introduction

Medical data from past decades have clearly shown that infertility is a growing challenge for public health, with increased numbers of reproductive failures. Currently, it is estimated that a male factor is present in half of infertility cases, but some data suggest that in the Middle East, the proportion of cases attributable to male factors is increased up to 70% [1,2,3]. In this aspect, varicocele, defined as an abnormal venous dilation of the pampiniform plexus, usually with blood reflux, is of exceptional importance because, according to European Association of Andrology (EAU) Guidelines from 2020, varicocele is the second most common cause of male infertility. In the general population, approximately 15% of adult men have varicoceles, but in the population of infertile men, the prevalence rises to 40% (primary infertility) or even to 80% (secondary infertility) [4,5,6,7,8]. Therefore, proper medical examination of the scrotum is essential for the management of male infertility [4,5,8,9,10,11,12,13,14,15,16,17].

The pathophysiology of varicocele is not completely explained. The hypotheses assume that the causes may include hypoplasia of the developing male gonad in adolescence, congenital/acquired valve defects, venous obstruction, and anatomical variations [8,18,19]. Varicocele-related infertility is an effect of changes in testicular blood flow, including blood reflux and venous pressure, leading to scrotal hyperthermia, testicular hypoxia, endocrine disturbance, reflux and accumulation of toxic metabolites of adrenal or renal origin and testicular atrophy [7,8,20,21,22,23]. Moreover, most of these pathologies are linked with abnormal spermatogenesis and testicular oxidative stress (OS) resulting from an imbalance between pro- and antioxidative processes. Extensive generation of reactive oxygen species (ROS) in the male reproductive system is especially harmful because it leads to increased membrane lipids, proteins and DNA oxidation. Defects of cellular macromolecules may lead to impaired sperm maturation and function, enhancement of sperm apoptosis and finally nuclear DNA strand breaks in male gametes. The latter disorders are of significant clinical importance for successful fertilization, embryo development and pregnancy establishment [7,14,22,23,24,25,26,27,28].

Recent studies show that ≤15% sperm DNA fragmentation (SDF) is a good predictor of male fertility associated with a high likelihood of achieving natural pregnancy; in turn, >15–30% SDF may be related to subinfertility and a reduced probability of reproductive success even if intrauterine insemination (IUI) is used, and >30% SDF is strongly associated with a high risk of male infertility with failure of fertilization, delayed or arrested embryo development and the inability to achieve pregnancy, either naturally or with assisted reproductive technology (ART). It should also be highlighted that >40% SDF is also associated with a significant increase in the occurrence of pregnancy loss and congenital defects in offspring [22,29,30,31,32,33,34].

Therefore, simultaneous assessment of standard semen parameters, OS in semen and SDF, has become clinically significant in varicocele-related infertility. This approach proves to be a better diagnostic and prognostic marker of male reproductive status than routine semen analysis alone [5,10,12,14,26,28]. For this reason, the aim of the study was to assess the influence of varicoceles not only on conventional semen characteristics but also on oxidation-reduction potential (ORP) and SDF. Furthermore, the relationships between oxidative stress in semen, sperm DNA damage and basic semen parameters were analysed.

## 2. Materials and Methods

### 2.1. Study Groups

The study was approved by The Ethics Committee of Pomeranian Medical University, Szczecin, Poland (ethical authorization number: KB-0012/21/2018). Consent for inclusion in the study was obtained from each subject. In the study population, 166 infertile men, 64 men with proven fertility and 105 healthy volunteers without any known health problems (of unknown fertility status) were enrolled. The latter two groups were considered as the controls. The Figureertile group consisted of men who had naturally produced offspring in the last 3 years or whose partners were pregnant during the recruitment of participants (Figure 1). Infertile men were selected from couples who were diagnosed/treated for infertility at the TFP Fertility Vitrolive in Szczecin (Poland). These subjects had not initiated a pregnancy within the past 12 months of regular, unprotected sexual intercourse. Seventy-one out of 166 infertile subjects had clinically diagnosed varicocele (varicocele-positive men) on Doppler ultrasonography (Mindray, China, type Z-5 with a linear head with a frequency range of 3–2.6 MHz). Unilateral varicocele was found in 70 patients, while bilateral varicocele was found in 1 patient. The varicocele was grade 1 (varicocele palpable during Valsava manoeuvre) in 9 patients, grade 2 (varicocele palpable at rest) in 33 patients, grade 3 (visible and palpable at rest) in 28 patients, and ungraded in 1 patient [6,26,35]. In 95 out of 166 infertile men, no varicoceles were found (varicocele-negative men).

The semen clinical categories were defined according to the criteria provided by the World Health Organization [36]. The healthy volunteers had normal standard semen variables (normozoospermic men). Normozoospermia was considered when the sperm concentration was ≥15 mln/mL, the total number of sperm cells was ≥39 mln per ejaculate, sperm progressive motility was ≥32%, and normal sperm morphology was ≥4%. In the fertile group, 45 men with normozoospermia, 1 with asthenozoospermia (low sperm motility), 5 with oligozoospermia (low sperm count), 1 with oligoteratozoospermia (coexistence of low sperm count and abnormal sperm morphology) and 12 with teratozoospermia (abnormal sperm morphology) were diagnosed. Most of the varicocele-positive infertile patients had reduced standard semen parameters: 5 men with asthenoteratozoospermia (coexistence of low sperm motility and abnormal sperm morphology), 1 with oligozoospermia, 16 with oligoteratozoospermia, 24 with oligoasthenoteratozoospermia (coexistence of low sperm count, low motility and abnormal sperm morphology) and 21 with teratozoospermia; only 4 men had normozoospermia. Varicocele-negative infertile men also presented variable basic sperm parameters: 11 with asthenoteratozoospermia, 3 with oligozoospermia, 17 with oligoteratozoospermia, 30 with oligoasthenoteratozoospermia, 14 with teratozoospermia and 20 with normozoospermia. All participants of the study were evaluated by medical history, physical examinations and semen analysis. The exclusion criteria were azoospermia; hypogonadism (testicular volume < 15 mL); a history of testicular torsion or atrophy; excess cigarette, alcohol or drug consumption; mumps; maldescent testis; injury or cancer; coexisting systemic and endocrine disease; current radiochemotherapy; exposure to gonadotoxins and leukocytospermia.

### 2.2. Manual Semen Analysis 

All participants reported to the Laboratory of Andrology in the Department of Histology and Developmental Biology (Pomeranian Medical University in Szczecin, Szczecin, Poland) for seminological analysis. Material for the study was collected between 2018 and 2021. Conventional semen analysis was carried out according to the guidelines provided by WHO, 2010 [36]. Semen was obtained by masturbation after 2–7 days of recommended sexual abstinence, and semen samples were allowed to liquefy for 30 min at 37 °C before analysis. Semen evaluation was performed within 1 h of ejaculation. Sperm concentration was analysed in an improved Neubauer haemocytometer (Heinz Hernez Medizinalbedarf GmbH, Hamburg, Germany), and sperm progressive and nonprogressive motility and sperm vitality (eosin-positive cells and hypoosmotic-reactive cells: HOS test-positive cells) were assessed under a phase-contrast microscope using a 40× objective (DM 500, Leica, Heerbrugg, Switzerland). In turn, native sperm smears were fixed and stained according to the Papanicolaou method (Aqua-Med, Lodz, Poland) to evaluate spermatozoal morphology (including the teratozoospermia index reflecting multiple morphological defects per spermatozoon—TZI) and were analysed under a bright light microscope (DM 500, Leica, Heerbrugg Switzerland) using a 100× objective oil immersion lens. The concentration of leukocytes in the semen samples (peroxidase-positive cells) was calculated using the Endtz test (LeucoScreen kit, FertiPro N.V., Beernem, Belgium) and assessed in an improved Neubauer haemocytometer.

### 2.3. Sperm Chromatin Dispersion (SCD) Test (Halosperm Test)

Evaluation of sperm DNA fragmentation (SDF) was carried out by the SCD test using a Halosperm G2 kit (Halotech DNA, Madrid, Spain) according to the manufacturer’s guidelines, including denaturation, lysis, dehydration, and staining of sperm cells with eosin and thiazine. The procedure was made up of the following steps: (1) agarose was melted using a hot water bath (95–100 °C) for 5 min and then maintained at 37 °C; (2) the sperm sample was diluted in PBS (phosphate buffered saline) to a maximum of 20 million sperm per milliliter; (3) 50 mL of the sperm sample was transferred to the Eppendorf tube with 100 mL agarose at 37 °C and mixed gently with a micropipette; (4) a drop of 8 μL of cell suspension was placed onto the centre of a super-coated slide and covered with a coverslip; (5) the slide was transferred into the refrigerator at 4 °C for 5 min to solidify the agarose; (6) the coverslip was gently removed by sliding it off. From this step, processing was performed at room temperature (22 °C); (7) a denaturant agent was applied to the fully immersed reaction area and removed after 7 min; (8) the lysis solution was applied to the fully immersed reaction area and removed after 20 min; (9) the slide was washed for 5 min and covered with abundant distilled water. After that, the sample was dehydrated by flooding with 70% ethanol for 2 min, followed by flooding with 100% ethanol for 2 min; (10) the dried slide was stained by eosin staining solution A for 7 min; and (11) after removing the slide, eosin dye was applied with thiazine staining solution B for 7 min.

#### Scoring Criteria

A minimum of 300 sperm cells per sample were counted under the 100× objective of a bright light microscope (DM 500, Leica Switzerland) according to the following criteria: (1) sperm cells without DNA fragmentation (spermatozoa with a large halo—those whose halo width was similar or higher than the diameter of the core or spermatozoa with a medium-sized halo greater than 1/3 of the diameter of the core), and (2) sperm cells with fragmented DNA (spermatozoa with a small halo—the halo width was similar to or smaller than 1/3 of the diameter of the core, and spermatozoa without a halo or without a halo and degraded chromatin—those that showed no halo and presented a core irregularly or were weakly stained) (Figure 2). The results are presented as the sum of spermatozoa with small or no halos and degraded spermatozoa divided by the total number of assessed sperm cells and multiplied by 100%.

### 2.4. Measurement of Oxidation-Reduction Potential in Semen

Oxidative stress was verified by measuring oxidation-reduction potential (ORP) in semen using the Male Infertility Oxidative System (MiOXSYS^®^, Aytu BioScience, Englewood, CO, USA) according to the manufacturer’s guidelines. After semen liquefaction, 30 μL of the sample was dropped into the sample port of the disposable MiOXSYS sensor and inserted into the MiOXSYS analyser. Data for ORP were expressed as mV/10^6^ sperm/mL. A high ORP value (>1.37 mV/10^6^ sperm/mL) indicated a current imbalance between pro- and antioxidant processes in the seminal ejaculate and was considered oxidative stress (OS) [37].

### 2.5. Data Analysis

The quantitative variables (standard semen parameters, SDF and ORP) were expressed as the mean ± standard deviation (SD) and median (range), and categorical data were expressed as the percentage. The Shapiro-Wilk test showed that the data were not normally distributed. Therefore, a nonparametric Kruskal-Wallis test was applied to compare quantitative variables between all male infertility categories and control groups. The categorical data were tested by the chi^2^ test. The Spearman rank correlation coefficient (r_s_) was used to describe the relationships between SDF, ORP and conventional sperm characteristics. Odds ratios (ORs) with 95% CIs (95% confidential intervals) for having a low, moderate and high level of SDF or high ORP value in all male categories were calculated. To interpret the strength dependence between parameters, the following levels of correlation were presumed: <0.2—lack of linear dependence, 0.2–0.4—weak dependence, >0.4–0.7—moderate dependence, >0.7–0.9—strong dependence, and >0.9—very strong dependence. A *p* value less than 0.05 was considered significant. Data analysis was performed using Statistica version 13.3 (StatSoft, Krakow, Poland) and MedCalc version 18.2.1 (MedCalc Software, Ostend, Belgium).

## 3. Results

### 3.1. Age and Standard Semen Parameters

The present study showed the differences in age and sperm conventional characteristics between study groups (Table 1). The median (range) age in years of healthy volunteers with normozoospermia was 28.00 (20.00–44.00), that of fertile men was 32.00 (22.00–47.00), that of infertile men with varicoceles was 32.00 (22.00–48.00), and that of infertile men without varicocele was 34.00 (24.00–49.00). Significance was noted between normozoospermia and varicocele-positive and varicocele-negative infertile groups and fertile men. There was no significant difference in age between the infertile study groups.

Lower standard sperm parameters (sperm concentration, total sperm count—total sperm per ejaculate, sperm morphology, motility and vitality) were found in varicocele-positive and varicocele-negative men than in healthy volunteers and men with proven fertility (Table 1). Moreover, a higher value of TZI was found in infertile subjects than in healthy volunteers. Additionally, varicocele-negative men had significantly higher TZI than fertile men. A higher concentration of leukocytes was noted in the infertile groups than in the healthy volunteers. The two infertile groups differed only in TZI, which was lower in varicocele-positive men. Normozoospermic men and men with proven fertility did not differ in standard semen parameters except for semen volume and TZI (Figure 1, Table 1).

### 3.2. Sperm DNA Dispersion

The percentage of SDF was significantly higher in varicocele-positive (median: 20.00%) and varicocele-negative (median: 18.00%) infertile men than in subjects with normozoospermia (median: 12.00%) and with proven fertility (median: 13.00%). There was no difference in SDF between the two infertile groups or between normozoospermic and fertile individuals (Figure 1 and Table 1).

According to other researchers [22,29,30,31,32,33,34], the prevalence among subjects of an SDF ≤15% (low level of nuclear DNA damage—high fertility status), ˃15–30% (moderate level of nuclear DNA damage—moderate fertility status) and an SDF >30% (high level of nuclear DNA damage—low fertility status) was assessed in the total group (n = 335) and separately in varicocele-positive men (n = 71), varicocele-negative men (n = 95), healthy volunteers (n = 105) and fertile men (n = 64). A lower percentage of men with low SDF levels and a higher percentage of men with high SDF levels were found in varicocele-positive and varicocele-negative men vs. control groups. A higher prevalence of moderate SDF levels was shown in infertile groups vs. healthy volunteers but not vs. proven fertile men. The infertile groups differed in the distribution of low and high sperm DNA damage. A higher proportion of men with ≤15% SDF and a lower proportion with >30% SDF were noted in varicocele-positive infertile men. There were no differences in the frequency of low, moderate and high SDF levels between the control groups (Figure 1 and Table 2).

The infertile groups had a significantly lower OR for low SDF level (OR = 0.1357, 0.2353, 0.0500, and 0.0867, respectively) and a significantly higher OR for high SDF level (OR = 57.8850, 16.8750, 118.3659, and 35.6721, respectively) than the control groups. A higher OR for moderate SDF levels (OR = 2.9714, 3.0773) was revealed in infertile groups vs. healthy volunteers only. Infertile men with varicocele had greater odds of a low SDF level (OR = 2.7124) and lower odds of a high SDF level (OR = 0.4806) than infertile men without varicocele. The odds of occurrence of low, moderate and high SDF levels in normozoospermic men and fertile subjects were not significantly different (Figure 1 and Table 3).

### 3.3. Oxidation-Reduction Potential in Semen

The ORP value was significantly higher in varicocele-positive (median: 3.99 mV/10^6^/mL) and varicocele-negative (median: 7.23 mV/10^6^/mL) infertile men than in subjects with normozoospermia (median: 1.29 mV/10^6^/mL) and in men with proven fertility (median: 0.81 mV/10^6^/mL) (Figure 1 and Table 1).

The distribution of high ORP values in the semen (>1.37 mV/10^6^/mL) was assessed in the total group (n = 167) and separately in varicocele-positive men (n = 36), varicocele-negative men (n = 41), healthy volunteers (n = 38) and fertile men (n = 52). The number of men with high ORP was significantly higher in the infertile groups than in the control groups. There were no differences in the frequency of high ORP between infertile groups. On the other hand, a higher proportion of men with high ORP was observed in healthy volunteers than in the fertile group (Figure 1 and Table 4). The odds of occurrence of high ORP were higher in the infertile group than in the control groups (OR: 5.3968, 20.4000, 6.8889, and 26.0400, respectively). There were no significant differences in OR for high ORP when comparing varicocele-positive and varicocele-negative subjects, but a significant difference was found between normozoospermic men and fertile men. Healthy volunteers had higher odds for ORP >1.37 mV/10^6^/mL (OR = 3.7800) (Figure 1 and Table 5).

### 3.4. Correlations between Study Parameters

Sperm nuclear DNA damage and oxidation-reduction potential in semen in the total group were associated with conventional sperm parameters. SDF and ORP values were significantly negatively correlated with sperm concentration (r_s_ = −0.347 and r_s_ = −0.770, respectively), total sperm count (r_s_ = −0.299 and r_s_ = −0.683, respectively), sperm morphology (r_s_ = −0.488 and r_s_ = −0.566, respectively), sperm progressive motility (r_s_ = −0.555 and r_s_ = −0.546, respectively), total sperm motility (r_s_ = −0.524 and r_s_ = −0.588, respectively), eosin-negative spermatozoa (r_s_ = −0.560 and r_s_ = −0.473, respectively) and HOS test-positive spermatozoa (r_s_ = −0.483 and r_s_ = −0.266, respectively). Moreover, SDF and ORP values were positively correlated with TZI (r_s_ = 0.207 and r_s_ = 0.303, respectively). A significant positive correlation was found between SDF and ORP levels (r_s_ = 0.364) (Table 6).

## 4. Discussion

It should be emphasized that in our research, significant biomarkers of male infertility were applied to assess both the integrity of sperm nuclear DNA (SCD test) and the oxidative stress in semen (ORP measurement). The assessment of these two parameters is fully justified because one of the reasons for the decrease in the integrity of sperm DNA is oxidative stress in the male reproductive system. The SCD is a recommended diagnostic test, while currently the ORP measurement is recognized as a novel biomarker of male infertility and is also widely recommended [1,3,22]. In addition, in this study, an in-depth statistical analysis of the obtained data was performed, going beyond comparative analysis.

This study revealed associations between varicocele-mediated male infertility and routine semen analysis parameters, oxidation-reduction potential in semen, and sperm nuclear DNA fragmentation. As expected, our study clearly demonstrated a decrease in standard sperm characteristics associated with severe sperm nuclear DNA damage and oxidative stress in semen in groups of positive- and negative-varicocele men with respect to the normozoospermic and fertile control groups (Figure 1).

### 4.1. The Detrimental Effect of Varicoceles on Conventional Sperm Characteristics

First, our study clearly shows that varicocele-mediated infertility is related to a significant decrease in the efficiency of spermatogenesis. We found that males with venous dilation of the pampiniform plexus had lower sperm counts, percentages of normal sperm forms, and motile and vital gametes than both healthy subjects with normal basic semen profiles and with proven fertility. It should be stressed that most patients with varicocele (67 out of 71) had abnormal routine sperm characteristics. Moreover, varicocele-positive and varicocele-negative infertile patients had similar results in terms of conventional semen analysis. These data indicated that varicocele could be a significant factor in spermatogenesis failure. The obtained results are consistent with the findings of other researchers who also presented decreased standard semen characteristics in the varicocele group in comparison with normozoospermic and fertile subjects [24,28,29,38,39]. In contrast, some authors found differences only in selected parameters [26,40,41,42]. Jeremias et al. [26] revealed that the varicocele-positive group had significantly lower sperm progressive motility than normozoospermic volunteers. In turn, Pallotti et al. [41] showed that subjects with varicocele had lower sperm concentrations than men without varicocele, while Redmon et al. [42] only noted lower total motile sperm counts in the ejaculate.

### 4.2. The Detrimental Effect of Varicoceles on Nuclear Sperm DNA

It should be highlighted that conventional semen analysis is still essential for male factor infertility evaluation; however, it is not sensitive enough to detect subtle sperm defects that may interfere with patient fertility. For this reason, many researchers have proposed that the SDF level be used as an independent biomarker of sperm quality that may have better diagnostic and prognostic capabilities than standard sperm parameters [7,14,22,24,26,28]. Additionally, our findings confirm that fertility status is strongly linked with sperm chromatin maturity/integrity. We showed not only a significant difference in the percentage (median) of sperm cells with fragmented DNA in the ejaculate between infertile men (both with or without varicocele) and controls but also that in the infertile groups, the prevalence of high levels of DNA damage (>30% SDF) was significantly higher, and that of low levels of DNA damage (≤15% SDF) was significantly lower, than in controls. Furthermore, the odds ratio for >30% SDF in infertile groups was between 16.8 and 118.3 times as high as that in normozoospermic and fertile men, while the odds ratio for ≤15% SDF was 0.23 to 0.05 times as low. These results clearly show that if the level of SDF is >30%, the risk of male infertility is great and that SDF ≤ 15% is a good predictor of male fertility. Interestingly, our data did not display significant differences in SDF analysis between varicocele-positive and varicocele-negative men, but the obtained value of the SCD test could suggest that the sperm nuclear DNA of men with varicoceles was more susceptible to damage than the sperm chromatin of varicocele-negative infertile men. The median SDF proportion was higher in subjects with varicoceles than in subjects without varicoceles (20.00% vs. 18.00%, respectively).

In addition, the detrimental effect of varicoceles on nuclear sperm DNA was confirmed by other researchers in the most recent studies. In the group of varicocele-mediated infertility vs. groups of men with normozoospermia and proven fertility, a higher proportion of TUNEL-positive sperm cells (60.87% vs. 8.14%; 69.88% vs. 8.14%; 20.20% vs. 10.10%) [24,39], sperm with fragmented DNA as evaluated by comet assay (total sperm DNA fragmentation: 72.00% vs. 64.5%; double-stranded DNA: 64.20% vs. 52.90% and 53.00% vs. 45.00%; single-stranded DNA: 68.50% vs. 28.00%; total oxidative DNA fragmentation: 76.37% vs. 66.88%) [16,26], and sperm with fragmented DNA assessed by SCSA (16.73% vs. 9.83%; 29.90% vs. 56.56%) [28,29] were noted. Additionally, a comparison of men with varicocele and normal standard semen parameters or with varicocele and abnormal standard semen parameters and normozoospermic men with unknown fertility status (volunteers) showed significant differences in the results of the SCD test (21.22% vs. 11.40% and 43.78% vs. 11.40%, respectively) [40]. 

It should be emphasized that our current research is consistent with the results published previously, which provide clear evidence that infertility and abnormal sperm parameters are strongly related to sperm chromatin status. We have shown that infertile subjects demonstrated significantly higher levels of SDF than healthy normozoospermic subjects (median: 23.00% vs. 14.00%). In the infertile group, the prevalence and odds ratio of >30% SDF were significantly higher, while those of ≤15% SDF were significantly lower. Moreover, the analysis of the receiver operating characteristic (ROC) curve and area under the curve (AUC) showed that a level of 20% SDF (AUC = 0.785) was the cut-off point to distinguish these two populations of men, and the group of infertile men had an over 6.5-fold higher risk of >20% SDF than normozoospermic subjects [43]. Another study conducted on a group of men with abnormal standard semen parameters and a group with a normal profile of basic semen characteristics confirmed these findings [44]. The experimentally calculated threshold level of SDF to discriminate these two groups was 18% (AUC = 0.753). Subjects from the group with decreased semen quality had a significantly higher prevalence of levels of >18% SDF and an over 5-fold increased risk of >18% SDF. Moreover, comparison of potentially infertile men with teratozoospermia (<4% normal sperm morphology) and men without teratozoospermia revealed that morphological abnormalities of sperm cells are linked to abnormalities in chromatin structure [45]. Subjects with teratozoospermia had significantly higher SDF levels (median: 22.00% vs. 13.00%), and in this group, the incidence of >30% SDF was significantly higher, while the incidence of ≤15% SDF was significantly lower. Additionally, the odds ratio for >30% SDF was over 9.5 times higher, and the one for ≤15% SDF was over 0.2 times lower. Similar to the above study, ROC analysis provided the level of 18% SDF as a cut-off point (AUC = 0.783) for distinguishing the enrolled group without teratozoospermia from the group with teratozoospermia. Additionally, the group with <4% normal sperm morphology had a higher prevalence and over 4.6 times greater risk of >18% SDF. In this study, we found that conventional semen characteristics were negatively correlated with the fragmentation of sperm DNA.

### 4.3. The Detrimental Effect of Varicoceles on the Oxidation-Reduction Potential in Semen

It should be pointed out that oxidative stress is a major common factor identified in male infertility, especially in men with varicocele [21,24,28,29,46,47]. In our studies, we found that the infertile groups had significantly imbalanced oxidation-reduction potential in the ejaculate. Both varicocele-positive and varicocele-negative infertile males had significantly higher levels of ORP than those in the two control groups. Furthermore, the median values of ORP in varicocele-positive (3.99 mV/10^6^ sperm cells/mL) and varicocele-negative men (7.23 mV/10^6^ sperm cells/mL) were above 1.37 mV/10^6^ sperm cells/mL, which is considered a threshold of oxidative stress in semen [37], while the median values of ORP in the controls were below this threshold. We also estimated that in both infertile groups, the frequency of subjects with OS in semen was significantly higher than in the controls, and the risk of OS in the semen of infertile males was over 5.3 times to over 26 times as high as in normozoospermic and fertile men. Moreover, the results of ORP assessment were negatively correlated with standard semen parameters and positively correlated with SDF levels (r_s_ = 0.364), suggesting that OS could be engaged in spermatogenesis failure and male gamete damage and ultimately lead to a decrease in male fertility. Our obtained results may suggest oxidative stress-induced sperm nuclear DNA fragmentation in men with varicoceles. This hypothesis is confirmed by other researchers. However, to the best of our knowledge, the number of scientific reports is limited. A higher level of ORP was also found in varicocele-positive infertile subjects than in healthy normozoospermic men (4.02 mV/10^6^ sperm cells/mL vs. 1.14 mV/10^6^ sperm cells/mL) [28]. Additionally, there were significant differences between the varicocele-associated infertility group and normozoospermic fertile controls in the levels of ROS (4.49 photons/min vs. 2.62 photons/min) and total antioxidant capacity (TAC) (0.97 mM vs. 1.50 mM) [29]. Furthermore, the authors revealed a positive correlation between levels of ORP and DFI (r = 0.320) [28], DFI and TAC (r = −0.669), between ROS level and DFI (r = 0.654) and between ROS level and TAC (r = −0.791) in the varicocele-positive group [29]. Additionally, sperm DNA fragmentation was positively correlated with the level of malondialdehyde (MDA—biomarker of lipid peroxidation: r = 0.735) and negatively correlated with the level of superoxide dismutase (r = −0.781), catalase (r = −0.686) and glutathione peroxidase (r = −0.721) in the total study group (men with or without varicoceles) [24].

## 5. Limitations of the Study

Some relevant limitations of our research should be noted. Firstly, the group of normozoospermic volunteers had unknown fertility status. To the best of our knowledge, they presented high fertility potential, which was reflected in comparison with the group of men with proven fertility; however, we cannot rule out the possibility that some individual subjects with reduced fertility were enrolled in this group. Moreover, in the group of men with proven fertility, we included subjects who became fathers in the last 3 years. We cannot exclude the possibility that in the time from fertilization to time of semen analysis in limited cases there were no harmful incidents influencing the obtained results. Finally, the varicocele-mediated group was heterogeneous because not all subjects had the same grade of varicocele; however, the vast majority had varicocele diagnosed as grade 2 or 3. Additionally, we cannot exclude possibility of some single false negative diagnoses of varicocele in the group of varicocele-negative infertile men, but we would like to point out that infertile men were examined by experienced urologists, and we believed that the risk of wrong diagnosis seems to be relatively low.

## 6. Conclusions

Based on the presented data, we can suggest that in infertile men, including varicocele-mediated cases, abnormal standard sperm characteristics coexist with a significant increase in sperm nuclear DNA fragmentation. Oxidative stress could also play a key role in this issue. Therefore, the diagnosis and treatment of male infertility should be focused not only on conventional semen assessment but also on verification of sperm chromatin status and identification and elimination of unbalanced ORP in semen. Our demonstrated findings revealed the close relationships between these parameters (Figure 1). Therefore, identification of the conditions or factors associated with sperm chromatin damage and oxidative stress is very important. It should be highlighted that some factors can be eliminated, while diagnostic parameters can be modified using proper medical management based on examination and in-depth medical interviews. Recent studies have shown that modification of unhealthy habits (e.g., smoking, alcohol abuse, drugs, and being overweight), administration of antioxidant therapy, and varicocelectomy in cases of varicocele-mediated infertility can contribute to restoring physiological levels of ROS and to improving semen quality, including the integrity of sperm DNA [8,9,10,12,13,14,15,16,21,29,48]. Moreover, in the era of growing numbers of chronic disease cases (e.g., insulin resistance, cancers, cardiovascular disease, and being overweight), it cannot be ignored that the adoption of healthy habits can additionally improve general male health and may contribute to a lower cost of medical care in the future.

## Figures and Tables

**Figure 1 ijerph-18-05977-f001:**
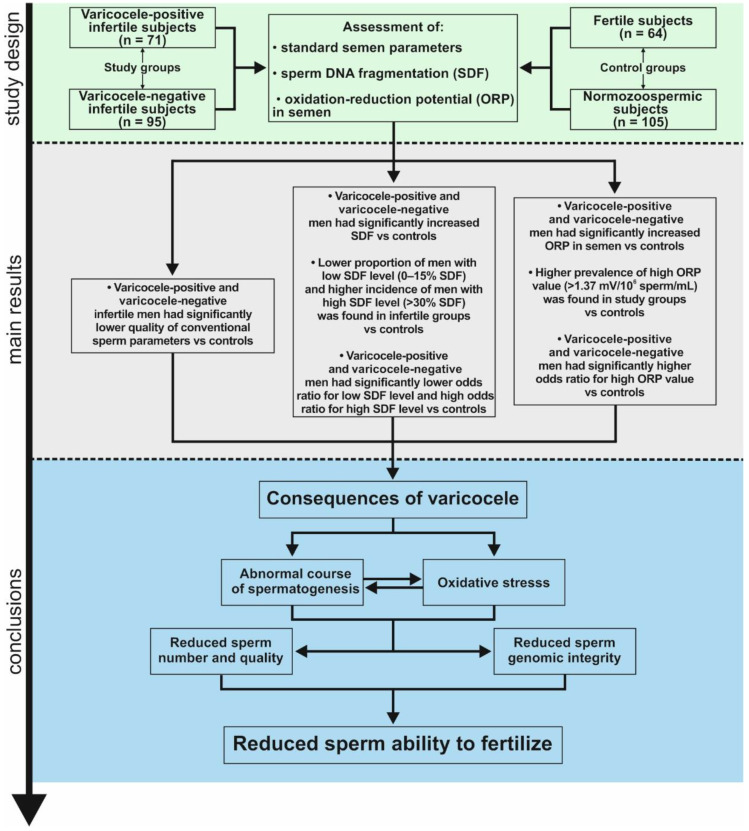
Study design, results and suggested pathomechanism associated with abnormalities of spermatogenesis in varicocele subjects.

**Figure 2 ijerph-18-05977-f002:**
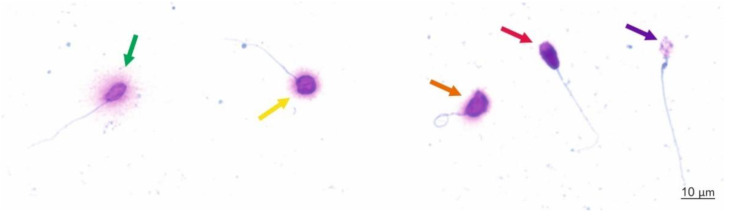
Micrograph of sperm chromatin dispersion test (SCD) results obtained by light microscopy. Sperm cells with large halos (green arrow) and medium-sized halos (yellow arrow) have normal integrity of nuclear DNA, and sperm cells with small halos (orange arrow), without halos (red arrow) or without halos and degraded chromatin (violet arrow) have fragmented nuclear DNA.

**Table 1 ijerph-18-05977-t001:** Descriptive statistics and comparisons of age, standard semen parameters, SDF and ORP between study groups.

Parameters	Total (n = 335)	Varicocele Positive-Infertile Men (1) (n = 71)	Varicocele Negative-Infertile Men (2) (n = 95)	Healthy Volunteers with Normozoospermia (3) (n = 105)	Men with Proven Fertility (4) (n = 64)	*p* 1 vs. 2	*p* 1 vs. 3	*p* 1 vs. 4	*p* 2 vs. 3	*p* 2 vs. 4	*p* 3 vs. 4
**Age (y)**	32.00 (20.00–49.00) 31.84 ± 5.66	32.00 (22.00–48.00) 32.61 ± 5.02	34.00 (24.00–49.00) 34.37 ± 5.37	28.00 (20.00–44.00) 28.80 ± 5.23	32.00 (22.00–47.00) 32.39 ± 5.13	NS	*p* = 0.000038	NS	*p* < 0.000001	NS	*p* = 0.000462
**Semen volume (mL)**	3.00 (0.50–10.00) 3.47 ± 1.53	3.00 (1.00–9.00) 3.43 ± 1.68	3.00 (0.50–8.00) 3.19 ± 1.43	3.50 (0.75–10.00) 3.89 ± 1.63	3.00 (0.50–8.00) 3.25 ± 1.74	NS	NS	NS	*p* = 0.011929	NS	*p* = 0.012485
**Sperm concentration (×10^6^/mL)**	25.38 (0.05–146.50) 29.16 ± 25.59	8.90 (0.25–146.50) 17.85 ± 23.90	10.50 (0.05–104.50) 17.39 ± 19.84	32.00 (8.25–119.50) 38.36 ± 21.61	33.82 (4.80–115.25) 44.10 ± 27.52	NS	*p* < 0.000001	*p* < 0.000001	*p* < 0.000001	*p* < 0.000001	NS
**Total number of spermatozoa (×10^6^)**	69.60 (0.25–672.00) 97.80 ± 96.43	29.97 (0.50–293.00) 55.57 ± 65.61	31.60 (0.25–412.75) 53.83 ± 69.24	117.08 (39.00–475.00) 141.70 ± 87.31	106.63 (21.60–672.00) 137.91 ± 122.51	NS	*p* < 0.000001	*p* < 0.000001	*p* < 0.000001	*p* < 0.000001	NS
**Morphologically normal spermatozoa (%)**	4.00 (0.00–13.00) 3.94 ± 3.39	1.00 (0.00–7.00) 1.26 ± 1.63	0.00 (0.00–11.00) 1.89 ± 2.79	7.00 (4.00–12.00) 6.68 ± 1.99	5.00 (0.00–13.00) 5.48 ± 3.24	NS	*p* < 0.000001	*p* < 0.000001	*p* < 0.000001	*p* < 0.000001	NS
**TZI**	1.68 (1.22–2.52) 1.72 ± 0.28	1.67 (1.35–2.50) 1.73 ± 0.24	1.94 (1.44–2.52) 1.95 ± 0.27	1.48 (1.22–2.12) 1.52 ± 0.18	1.70 (1.32–2.25) 1.69 ± 0.19	*p* = 0.000017	*p* = 0.000001	NS	*p* < 0.000001	*p* = 0.000003	*p* = 0.000029
**Progressive motility (%)**	59.00 (0.00–94.00) 53.11 ± 23.07	39.00 (0.00–85.00) 36.70 ± 19.92	38.00 (00.00–84.00) 39.54 ± 23.53	69.00 (34.00–94.00) 68.54 ± 11.24	68.50 (22.00–90.00) 66.12 ± 13.99	NS	*p* < 0.000001	*p* < 0.000001	*p* < 0.000001	*p* < 0.000001	NS
**Nonprogressive motility (%)**	6.00 (0.00–29.00) 6.99 ± 4.63	6.00 (0.00–16.00) 6.21 ± 3.83	5.00 (0.00–29.00) 5.66 ± 4.09	8.00 (0.00–18.00) 8.03 ± 4.15	6.00 (1.00–26.00) 8.12 ± 6.10	NS	*p* = 0.030324	NS	*p* = 0.000087	NS	NS
**Total sperm motility (%)**	67.00 (0.00–98.00) 60.10 ± 23.77	46,00 (0.00–87.00) 42.91 ± 20.44	45.00 (0.00–87.00) 45.21 ± 23.60	79.00 (52.00–98.00) 76.58 ± 10.13	77.00 (28.00–98.00) 74.25 ± 14.39	NS	*p* < 0.000001	*p* < 0.000001	*p* < 0.000001	*p* < 0.000001	NS
**Eosin-negative spermatozoa—live cells (%)**	80.00 (3.00–98.00) 76.76 ± 14.38	72.00 (3.00–91.00) 68.42 ± 17.19	76.00 (23.00–96.00) 72.29 ± 15.26	84.00 (60.00–98.00) 82.64 ± 8.95	85.00 (48.00–98.00) 83.00 ± 9.11	NS	*p* < 0.000001	*p* < 0.000001	*p* = 0.000001	*p* = 0.000006	NS
**HOS test-positive spermatozoa—live cells (%)**	n = 291 80.00 (13.00–98.00) 76.68 ± 13.39	n = 49 71.00 (13.00–90.00) 68.51 ± 16.59	n = 77 73.00 (20.00–92.00) 71.96 ± 15.05	n = 105 85.00 (58.00–94.00) 81.68 ± 8.81	n = 60 82.00 (50.00–98.00) 80.66 ± 9.13	NS	*p* < 0.000001	*p* = 0.000137	*p* = 0.000006	*p* = 0.003885	NS
**Peroxidase-positive cells (mln/mL)**	0.20 (0.00–0.96) 0.23 ± 0.24	0.25 (0.00–0.96) 0.32 ± 0.23	0.25 (0.00–0.90) 0.29 ± 0.23	0.00 (0.00–0.90) 0.13 ± 0.20	0.20 (0.00–0.95) 0.22 ± 0.24	NS	*p* < 0.000001	NS	*p* = 0.000002	NS	NS
**SDF (%)**	20.00 (2.00–64.00) 23.29 ± 11.89	20.00 (2.00–64.00) 23.29 ± 11.89	18.00 (4.00–53.00) 19.35 ± 9.54	12.00 (3.00–28.00) 13.33 ± 5.93	13.00 (3.00–34.00) 13.85 ± 7.13	NS	*p* < 0.000001	*p* < 0.000001	*p* = 0.000006	*p* = 0.000899	NS
**ORP (mV/10^6^ sperm/mL)**	n = 167 1.69 (0.02–196.50) 14.32 ± 38.17	n = 41 3.99 (0.28–196.50) 36.10 ± 60.97	n = 36 7.23 (0.68–169.11) 22.51 ± 40.43	n = 38 1.29 (0.29–4.98) 1.58 ± 0.91	n = 52 0.81 (0.02–4.38) 1.00 ± 0.80	NS	*p* = 0.0000657	*p* < 0.000001	*p* = 0.000147	*p* < 0.000001	NS

Data are expressed as median (range) and mean ± SD, HOS test—hypoosmotic swelling test, n—number of subjects, NS—not significant, ORP—oxidation-reduction potential in semen, SD—standard deviation, SDF—sperm DNA fragmentation, TZI—teratozoospermia index. Statistical significance in the Kruskal-Wallis test was reached when *p* < 0.05.

**Table 2 ijerph-18-05977-t002:** Prevalence of SDF in study groups.

Level of SDF (%)	Total (n = 335)	Varicocele Positive-Infertile Men (1) (n = 71) %(n)	Varicocele Negative-Infertile Men (2) (n = 95) %(n)	Healthy Volunteers with Normozoospermia (3) (n = 105) %(n)	Men with Proven Fertility (4) (n = 64) %(n)	*p* 1 vs. 2	*p* 1 vs. 3	*p* 1 vs. 4	*p* 2 vs. 3	*p* 2 vs. 4	*p* 3 vs. 4
**>30**	8.36(28)	21.13(15)	35.79(34)	0(0)	1.56(1)	*p* = 0.0411	*p* < 0.0001	*p* = 0.005	*p* < 0.0001	*p* < 0.0001	NS
**>15–30**	40.30(135)	50.70(36)	51.58(49)	25.71(27)	35.94(23)	NS	*p* = 0.0057	NS	*p* < 0.0001	NS	NS
**≤15**	51.34(172)	28.17(20)	12.63(12)	74.29(78)	62.50(40)	*p* = 0.0123	*p* < 0.0001	*p* = 0.0001	*p* < 0.0001	*p* < 0.0001	NS

n—number of subjects, NS—not significant, SDF—sperm DNA fragmentation. Statistical significance in the chi^2^ test was reached when *p* < 0.05.

**Table 3 ijerph-18-05977-t003:** Odds ratio for SDF in study groups.

Level of SDF (%)	Varicocele Positive-Infertile Men (n = 71) %(n)	Varicocele Negative-Infertile Men (n = 95) %(n)	Healthy Volunteers with Normozoospermia (n = 105) %(n)	Men with Proven Fertility (n = 64) %(n)	OR1 (95% CI) *p*	OR2 (95% CI) *p*	OR3 (95% CI) *p*	OR4 (95% CI) *p*	OR5 (95% CI) *p*	OR6 (95% CI) *p*
**>30%**	21.13(15)	35.79(34)	0(0)	1.56(1)	0.4806 (0.2368–0.9751) *p* = 0.0424	57.8850 (3.40 –985.50) *p* = 0.0050	16.8750 (2.16–131.88) *p* = 0.0071	118.3659 (7.13–1965.11) *p* = 0.0009	35.6721 (4.74–268.72) *p* = 0.0005	0.2006 (0.01–5.00) NS
**>15–30%**	50.70(36)	51.58(49)	25.71(27)	35.94(23)	0.9656 (0.52–1.78) NS	2.9714 (1.57–5.63) *p* = 0.0008	1.8335 (0.92–3.66) NS	3.0773 (1.70–5.58) *p* = 0.0002	1.8989 (0.99–3.64) NS	0.6171 (0.32–1.21) NS
**≤15%**	28.17(20)	12.63(12)	74.29(78)	62.50(40)	2.7124 (1.22–6.01) *p* = 0.0140	0.1357 (0.07–0.27) *p* < 0.0001	0.2353 (0.11–0.49) *p* = 0.0001	0.0500 (0.02–0.11) *p* < 0.0001	0.0867 (0.04–0.19) *p* < 0.0001	1.7333 (0.89–3.38) NS

95% CI—95% confidential interval, n—number of subjects, NS—non statistically significant, OR—odds ratio, OR1—OR for SDF in infertile men with varicocele vs. infertile men without varicocele, OR2—OR for SDF in infertile men with varicocele vs. healthy volunteers, OR3—OR for SDF in infertile men with varicocele vs. fertile men, OR4—OR for SDF in infertile men without varicocele vs. healthy volunteers, OR5—OR for SDF in infertile men without varicocele vs. fertile men, OR6—OR for SDF in healthy volunteers vs. fertile men. Statistical significance in odds ratio test was reached when *p* < 0.05.

**Table 4 ijerph-18-05977-t004:** Prevalence of high ORP in study groups.

ORP	Total (n = 167)	Varicocele Positive-Infertile Men (1) (n = 41) %(n)	Varicocele Negative-Infertile Men (2) (n = 36) %(n)	Healthy Volunteers with Normozoospermia (3) (n = 38) %(n)	Men with Proven Fertility (n = 52) (4) %(n)	*p* 1 vs. 2	*p* 1 vs. 3	*p* 1 vs. 4	*p* 2 vs. 3	*p* 2 vs. 4	*p* 3 vs. 4
**>1.37 (mV/10^6^ sperm/mL)**	83.05(93)	82.92(34)	86.11(31)	47.37(18)	19.23(10)	NS	*p* = 0.0009	*p* < 0.0001	*p* = 0.0005	*p* < 0.0001	*p* = 0.0046

n—number of subjects, NS—not significant, ORP—oxidative-reduction potential. Statistical significance in the chi^2^ test was reached when *p* < 0.05.

**Table 5 ijerph-18-05977-t005:** Odds ratio for high ORP in study groups.

ORP	Varicocele Positive-Infertile Men (n = 41) %(n)	Varicocele negative-Infertile Men (n = 36) %(n)	Healthy Volunteers with Normozoospermia (n = 38) %(n)	Men with Proven Fertility (n = 52) %(n)	OR1 (95% CI) *p*	OR2 (95% CI) *p*	OR3 (95% CI) *p*	OR4 (95% CI) *p*	OR5 (95% CI) *p*	OR6 (95% CI) *p*
**>1.37 (mV/10^6^ sperm/mL)**	82.92(34)	86.11(31)	47.37(18)	19.23(10)	0.7834 (0.23–2.73) NS	5.3968 (1.92–15.16) *p* = 0.0014	20.4000 (7.02–59.27) *p* < 0.0001	6.8889 (2.21–21.52) *p* = 0.0009	26.0400 (8.0855–83.8637) *p* < 0.0001	3.7800 (1.48–9.66) *p* = 0.0055

95% CI—95% confidential interval, n—number of subjects, NS—non statistically significant, OR—odds ratio, OR1—OR for high ORP in infertile men with varicocele vs. infertile men without varicocele, OR2—OR for high ORP in infertile men with varicocele vs. healthy volunteers, OR3—OR for high ORP in infertile men with varicocele vs. fertile men, OR4—OR for high ORP in infertile men without varicocele vs. healthy volunteers, OR5—OR for high ORP in infertile men without varicocele vs. fertile men, OR6—OR for high ORP in healthy volunteers vs. fertile men, ORP—oxidation-reduction potential in semen. Statistical significance in the odds ratio test was reached when *p* < 0.05.

**Table 6 ijerph-18-05977-t006:** Rank Spearman correlation (r_s_) between SDF, ORP, male age and standard semen parameters in the total group.

Parametr	SDF (%) r_s_(*p*) (n = 335)	ORP (mV/10^6^ Sperm/mL) r_s_(*p*) (n = 167)
**Age (y)**	0.178(0.001025)	0.015(NS)
**Semen volume (mL)**	0.064(NS)	0.041(NS)
**Sperm concentration (×10^6^/mL)**	−0.347(<0.000001)	−0.770(<0.000001)
**Total number of spermatozoa (×10^6^)**	−0.299(<0.000001)	−0.683(<0.000001)
**Morphologically normal spermatozoa (%)**	−0.488(<0.000001)	−0.566(<0.000001)
**TZI**	0.207(0.0000126)	0.303(0.000068)
**Progressive motility (%)**	−0.555(<0.000001)	−0.546(0.000001)
**Nonprogressive motility (%)**	0.040(NS)	−0.275(0.000324)
**Total sperm motility (%)**	−0.524(<0.000001)	−0.588(<0.000001)
**Eosin-negative spermatozoa—live cells (%)**	−0.560(<0.000001)	−0.473(<0.000001)
**HOS test-positive spermatozoa—live cells (%)**	n = 291 −0.483(<0.000001)	n = 142 −0.266(0.001457)
**SDF (%)**	–	0.364(<0.000001)

HOS test—hypoosmotic swelling test, n—number of subjects, NS—non statistically significant, OPR—oxidation-reduction potential, SDF—sperm DNA fragmentation, TZI—teratozoospermia index. Statistical significance in the rank Spearman correlation was reached when *p* < 0.05. Interpretation of the r_s_ value is as follows: <0.2 lack of linear dependence, ≥0.2–0.4—weak dependence, >0.4–0.7—moderate dependence, >0.7–0.9—strong dependence, >0.9—very strong dependence.

## Data Availability

The data presented in this study are available on request from the corresponding author.

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
