# Peer review of "The Negative Impact of Varicocele on Basic Semen Parameters, Sperm Nuclear DNA Dispersion and Oxidation-Reduction Potential in Semen"

_ijerph, 2021, doi:10.3390/ijerph18115977_

Round 1

Reviewer 1 Report

The authors studied the negative impact of varicocele on sperm parameters, specifically sperm nuclear DNA dispersion and oxidation-reduction potential (ORP) in semen.

1- The introduction is very lengthy. It should be shortened and focused. Part of the introduction should go into the discussion section.

2- The material&Methods and results section are well written. However, a new analysis should be performed. The results should be stratified according to the varicocele grade. It is well known that mostly grade 2 and 3 varicoceles are clinically significant. It would be interesting to see the difference in semen parameters and DFI and ORP for the different grades.

3- In the discussion section:

  • A new section should be written regarding the limitation of the studies.
  • The authors should refrain from commenting on the fertility potential. The term "reproductive failure" should not be used. This article studied the effects of varicocele on semen parameters and not on ART outcomes or natural pregnancy rates. Therefore, the extrapolation should be eliminated. Semen parameters can be different from fertility potential and outcomes.
  • The authors should re-phrase some part in the discussion: this work only proves correlation and association and not causality. 
  • One example among many: "we can assume that in male infertility, including varicocele-mediated cases, abnormal standard sperm characteristics and a significant increase in sperm nuclear DNA fragmentation could be responsible for reproductive failure." The authors should not assume, and they don't have enough data to support the assumption. They should only report the negative correlation they observed. 
  • Some English spelling mistakes need to be addressed. 

Author Response

Dear Reviewer,

Thank you for your all suggestions and recommendations. We agree with your valuable suggestions. All requested changes have been marked in red colour and performed in the revised version of the manuscript. Detailed responses to your suggestions and recommendations are listed below.

The authors studied the negative impact of varicocele on sperm parameters, specifically sperm nuclear DNA dispersion and oxidation-reduction potential (ORP) in semen.

1- The introduction is very lengthy. It should be shortened and focused. Part of the introduction should go into the discussion section.

Response: We agree with you, some irrelevant sentences have been removed from the introduction of manuscript and this section has been shortened.

2- The material&Methods and results section are well written. However, a new analysis should be performed. The results should be stratified according to the varicocele grade. It is well known that mostly grade 2 and 3 varicoceles are clinically significant. It would be interesting to see the difference in semen parameters and DFI and ORP for the different grades.

Response: We agree with you that comparison of the groups with different degrees of varicocele may have great clinical importance. However, in our study separated groups with grade I, II, or III will be quite small, especially the grade I (only 9 subjects). We believe that comparative analysis performed on so small and unequal groups may result in uncertain statistical data. In the future, we plan to apply your suggestion and conduct research focused on groups with different degree of varicocele. Moreover, the aim of this study was to compare the changes in semen parameters, SDF and oxidative stress between infertile men with varicocele, infertile subjects without varicocele as well as healthy volunteers and men with proven fertility.

3- In the discussion section:

  • A new section should be written regarding the limitation of the studies.

Response: We agree with you, the new section is added.

  • The authors should refrain from commenting on the fertility potential. The term "reproductive failure" should not be used. This article studied the effects of varicocele on semen parameters and not on ART outcomes or natural pregnancy rates. Therefore, the extrapolation should be eliminated. Semen parameters can be different from fertility potential and outcomes.

Response: We agree with you, the controversial sentences in the manuscript were removed and some sentences were rewritten. We focused on negative effects of varicocele on semen parameters (standard parameters, sperm DNA fragmentation and value of oxidation-reduction potential). Moreover, we completely agree with you that semen parameters can be different from fertility potential and outcomes what we mentioned in discussion: ‘For this reason, many researchers have proposed that the SDF level be used as an independent biomarker of sperm quality that may have better diagnostic and prognostic capabilities than standard sperm parameters’    

  • The authors should re-phrase some part in the discussion: this work only proves correlation and association and not causality. 

Response: We agree with you, section ‘Concluding remarks and comments’ has been improved and sentences suggesting causality have been corrected.

  • One example among many: "we can assume that in male infertility, including varicocele-mediated cases, abnormal standard sperm characteristics and a significant increase in sperm nuclear DNA fragmentation could be responsible for reproductive failure." The authors should not assume, and they don't have enough data to support the assumption. They should only report the negative correlation they observed. 

Response: We agree with you, the manuscript has been improved. Now we more highlighted and focused on the negative associations between evaluated parameters.

  • Some English spelling mistakes need to be addressed. 

Response: We apologize for the mistakes, but the manuscript was checked by a professional language editing company. We introduced all the changes proposed by a native speaker and received a language certificate from the American Journal Experts that we sent to the International Journal of Environmental Research and Public Health during submission of the manuscript. However,  we re-checked and improved the manuscript.

Reviewer 2 Report

In this paper, Kamil Gill et al. report that semen from varicocele-positive and negative infertile men exhibit lower standard sperm parameters and higher sperm DNA fragmentation (SDF) and oxidation-reduction potential (ORP), compared with semen from fertile men and healthy volunteers. Furthermore, the authors show a negative correlation between SDF/ORP and several standard sperm parameters, and a positive relationship between SDF and ORP.

The paper is well-written and the data are sound. However, the reviewer thinks that most of results described in the present paper have been addressed in prior publications.

e.g.,

  • Agarwal A, Sharma RK, Desai NR, Prabakaran S, Tavares A, Sabanegh E. Role of oxidative stress in pathogenesis of varicocele and infertility. Urology.2009:73:461-9
  • Altunoluk B, Efe E, Kurutas EB, Gul AB, Atalay F, Eren M. Elevation of both reactive oxygen species and antioxidant enzymes in vein tissue of infertile men with varicocele. Urol Int. 2012:88:102-6
  • Allamaneni SS, Naughton CK, Sharma RK, Thomas AJ, Jr., Agarwal A. Increased seminal reactive oxygen species levels in patients with varicoceles correlate with varicocele grade but not with testis size. Fertil Steril. 2004:82:1684-6
  • Pasqualotto FF, Sundaram A, Sharma RK, Borges E, Jr., Pasqualotto EB, Agarwal A. Semen quality and oxidative stress scores in fertile and infertile patients with varicocele. Fertil Steril. 2008:89:602-7
  • Agarwal A, Hamada A, Esteves SC. Insight into oxidative stress in varicocele-associated male infertility: part 1. Nat Rev Urol. 2012:9:678-90
  • Abd-Elmoaty MA, Saleh R, Sharma R, Agarwal A. Increased levels of oxidants and reduced antioxidants in semen of infertile men with varicocele. Fertil Steril. 2010:94:1531-4
  • Aitken RJ, Curry BJ. Redox regulation of human sperm function: from the physiological control of sperm capacitation to the etiology of infertility and DNA damage in the germ line. Antioxid Redox Signal. 2011:14:367-81
  • Sakkas D, Alvarez JG. Sperm DNA fragmentation: mechanisms of origin, impact on reproductive outcome, and analysis. Fertil Steril. 2010:93:1027-36

The reviewer has no idea the distinction between the current paper and these previous reports. The authors should emphasize and explain the novelty of this paper in “Title” and “Discussion”.  

Author Response

Dear Reviewer,

Thank you for your review. We agree with your that novelty of the study should be more highlighted for that the title and discussion have been improved (red colour). We also agree with the fact that research on varicocele-mediated infertility has been conducted for many years. However, to the best of our knowledge, the number of scientific reports including simultaneous comparison of standard semen parameters, sperm chromatin status evaluation and oxidation-potential in semen between group of infertile men with varicocele, infertile men without varicocele, healthy volunteers and men with proven fertility is not widely described. Moreover, studies presented statistical analysis of prevalence and odds ratio of different levels of sperm DNA fragmentation and oxidative stress in the mentioned groups are limited. For that we believe that the presented results provide new data about negative influence of varicocele on the semen parameters. In addition, below we would like to highlight the main differences between our research and the publications of other authors.

In this paper, Kamil Gill et al. report that semen from varicocele-positive and negative infertile men exhibit lower standard sperm parameters and higher sperm DNA fragmentation (SDF) and oxidation-reduction potential (ORP), compared with semen from fertile men and healthy volunteers. Furthermore, the authors show a negative correlation between SDF/ORP and several standard sperm parameters, and a positive relationship between SDF and ORP.

The paper is well-written and the data are sound. However, the reviewer thinks that most of results described in the present paper have been addressed in prior publications.

e.g.,

  • Agarwal A, Sharma RK, Desai NR, Prabakaran S, Tavares A, Sabanegh E. Role of oxidative stress in pathogenesis of varicocele and infertility. 2009:73:461-9

Response: This is a valuable review focusing on the role of oxidative stress in semen, while our manuscript is an original paper showing the relationship between varicocele, basic semological parameters, sperm nuclear DNA fragmentation as well as oxidation-reduction potential in semen. We focused on the analysis and comparison of the studied parameters between the infertile men with varicocele infertile men without varicocele, healthy volunteers and fertile men. Moreover we calculated prevalence of different levels of sperm DNA fragmentation and oxidative stress in semen. We also estimated odds ratio for distinguished levels of sperm DNA fragmentation and oxidative stress in semen in group of men with varicocele in comparison with varicocele-negative infertile men, healthy volunteers and fertile men.

  • Altunoluk B, Efe E, Kurutas EB, Gul AB, Atalay F, Eren M. Elevation of both reactive oxygen species and antioxidant enzymes in vein tissue of infertile men with varicocele. Urol Int. 2012:88:102-6

Response: This original study compared markers of oxidative stress in spermatic vein in men with varicocele and inguinal hernia. It does not include the assessment of semological parameters (standard semen parameters, sperm DNA fragmentation and oxidation-reduction potential in semen) which was carried out in our work. We believe that the idea and aim of Altunoluk et al. work was different from ours.

  • Allamaneni SS, Naughton CK, Sharma RK, Thomas AJ, Jr., Agarwal A. Increased seminal reactive oxygen species levels in patients with varicoceles correlate with varicocele grade but not with testis size. Fertil Steril. 2004:82:1684-6

Response: This original study was focused on the comparison of ROS levels in men with different grades of varicocele, but did not include the assessment of standard semen parameters and sperm DNA fragmentation as well as did not include the comparison of the results of men with varicoceles to control groups. We believe that the results presented in our manuscript do not duplicate the data presented by Allamaneni et al.

  • Pasqualotto FF, Sundaram A, Sharma RK, Borges E, Jr., Pasqualotto EB, Agarwal A. Semen quality and oxidative stress scores in fertile and infertile patients with varicocele. Fertil Steril. 2008:89:602-7

Response: In this publication, like in our study, authors compared group of infertile men with varicocele, healthy volunteers and fertile men, but sperm nuclear DNA fragmentation was not assessed, and the statistics include only comparison group. Moreover,  the size of the groups was smaller than in our manuscript, the standard semen parameters were assessed according to the previous WHO guidelines from 1999 (in our manuscript semen analysis was performed according WHO 2010) and oxidative stress was evaluated in different way (we used diagnostic system – Male Infertility Oxidative System –MiOXSYS®, Aytu BioScience, Colorado, USA).

  • Agarwal A, Hamada A, Esteves SC. Insight into oxidative stress in varicocele-associated male infertility: part 1. Nat Rev Urol. 2012:9:678-90

Response: It is a very valuable publication focusing on the pathomechanism dependent on oxidative stress in varicocele cases. However, this is a review paper and does not provide original data about decreased basic semen characteristic, increased sperm nuclear DNA fragmentation and oxidative stress in infertile men with varicocele, without varicocele and in group of men with high fertility potential.

  • Abd-Elmoaty MA, Saleh R, Sharma R, Agarwal A. Increased levels of oxidants and reduced antioxidants in semen of infertile men with varicocele. Fertil Steril. 2010:94:1531-4

Response: In this publication, as in our manuscript, infertile men with varicocele and fertile men were enrolled, but Abd-Elmoaty et al. focused on the assessment of the level of antioxidant enzymes and markers of oxidative stress in semen as well as on the correlation of these parameters with standard semen parameters carried out in according to WHO 1999 (in our study semen evaluation was performed according to WHO 2010). Moreover, Abd-Elmoaty et al did not evaluate sperm DNA fragmentation. Additionally, our study included varicocele-positive and -negative infertile men, healthy volunteers and men with proven fertility. The statistical analysis included the calculation of the frequency of different levels of SDF and ORP and the odds ratio of different levels of DNA damage and for the oxidative stress in semen of men with varicocele in relation to the other three enrolled groups.

  • Aitken RJ, Curry BJ. Redox regulation of human sperm function: from the physiological control of sperm capacitation to the etiology of infertility and DNA damage in the germ line. Antioxid Redox Signal. 2011:14:367-81

Response: Similarly as in the publication of Agarwal et al., this is also valuable paper. Aitken and Curry described the physiological and pathological role of reactive oxygen species in the regulating of sperm cells functions. However, due to the fact that this review publication, we believe that our manuscript is not a duplication of the results published by Aitken and Curry.

  • Sakkas D, Alvarez JG. Sperm DNA fragmentation: mechanisms of origin, impact on reproductive outcome, and analysis. Fertil Steril. 2010:93:1027-36

Response: As above, this is a very good review focusing on the cause of DNA damage in male reproductive cells. In truth Sakkas and Alvarez described the origin of sperm DNA fragmentation, but they did not focused on the varicocele issue, also they did not presented data similar to presented in our study.

The reviewer has no idea the distinction between the current paper and these previous reports. The authors should emphasize and explain the novelty of this paper in “Title” and “Discussion”.  

Response: We agree with your that novelty of the study should be more highlighted for that the title and discussion have been improved.

Reviewer 3 Report

I commend the authors for this very interesting manuscript that highlights de association between infertility and sperm DNA damage / oxidative stress. The authors clearly demonstrated an increase in semen oxidative stress and subsequent sperm DNA fragmentation in infertile men with or without varicocele. The authors also reported the moderate correlations between sperm DNA damage / oxidative stress and several sperm parameters. 

There are just a few points that deserve further clarification

  • The definition of grade 1 varicocele (dilated veins of <2.5 mm in diameter, with no flow reversal after Valsalva manoeuvre) is controversial. The 3 references cited by the authors don’t use this definition of varicocele, and most authors define varicocele as veins with diameter > 2,5mm and the presence of venous reflux. Thus, the authors should clarify why this definition was chosen, or exclude the 7 participants with grade 1 varicocele based on this definition
  • I suggest the author to assess the association between normal semen analysis parameters and sperm DNA fragmentation / oxidative stress. The authors clearly demonstrated that both parameters increase with decreased semen quality, however, it would be interesting to see how many men with normal semen analysis parameters in the infertility groups have increased sperm DNA fragmentation or ORP

Author Response

Dear Reviewer,

Thank you for your valuable and accurate suggestions and recommendations. Detailed responses to your suggestions are listed below.

I commend the authors for this very interesting manuscript that highlights de association between infertility and sperm DNA damage / oxidative stress. The authors clearly demonstrated an increase in semen oxidative stress and subsequent sperm DNA fragmentation in infertile men with or without varicocele. The authors also reported the moderate correlations between sperm DNA damage / oxidative stress and several sperm parameters. 

There are just a few points that deserve further clarification

  • The definition of grade 1 varicocele (dilated veins of <2.5 mm in diameter, with no flow reversal after Valsalva manoeuvre) is controversial. The 3 references cited by the authors don’t use this definition of varicocele, and most authors define varicocele as veins with diameter > 2,5mm and the presence of venous reflux. Thus, the authors should clarify why this definition was chosen, or exclude the 7 participants with grade 1 varicocele based on this definition

Response: Thank you for this comment. We completely agree with you that the definition should be clarify. We improved this part of manuscript according to the last EAU Guidelines (Salonia et al., 2020).

  • I suggest the author to assess the association between normal semen analysis parameters and sperm DNA fragmentation / oxidative stress. The authors clearly demonstrated that both parameters increase with decreased semen quality, however, it would be interesting to see how many men with normal semen analysis parameters in the infertility groups have increased sperm DNA fragmentation or ORP

Response: We agree with you that it is very interesting to assess the links between basic semen parameters and novel biomarkers of male fertility. In our previous original papers (cited in discussion of this manuscript) we focused on relationship between semen parameters (including sperm DNA fragmentation) in subjects with normal and abnormal standard semen parameters or in infertile subjects. The goal of this paper was different and provide new data. We wanted to verify association between varicocele-mediated infertility, standard semen parameters, sperm chromatin fragmentation and oxidation-reduction potential in semen. Therefore, we enrolled infertile men with varicocele and without varicocele and two groups with high fertility potential (young normozoospermic healthy volunteers and men with proven fertility). We also performed extensive statistical analysis of the obtained research results. On the other hand, in some part your idea has been realized, because we compared the group of men with only normal standard semen parameters (volunteers) and groups with men mostly presented abnormalities in basic semen parameters (two groups of infertile men).

Round 2

Reviewer 1 Report

The authors had adequate response to my comments and suggestions. 

Reviewer 2 Report

The second version of this manuscript has been revised well. The reviewer has no further comments.